# Experimental Study on Improving the Mechanical Properties of Material Extrusion Rapid Prototyping Polylactic Acid Parts by Applied Vibration

**Shijie Jiang [1,2], Tiankuo Dong [1,2,\*], Yang Zhan [3], Weibing Dai [1] and Ming Zhan [4]**

[1] School of Mechanical Engineering and Automation, Northeastern University, Shenyang 110819, China; jiangsj@me.neu.edu.cn (S.J.); 1810113@stu.neu.edu.cn (W.D.)

[2] Key Laboratory of Dynamics and Reliability of Mechanical Equipment of Liaoning Province, Northeastern University, Shenyang 110819, China

[3] Department of Cultural Foundation, Liaoning Guidaojiaotong Polytechnic Institute, Shenyang 110023, China; zhanming@ise.neu.edu.cn

[4] College of Information Science and Engineering, Northeastern University, Shenyang 110819, China; zhanming@mail.neu.edu.cn

\* Correspondence: 1870100@stu.neu.edu.cn; Tel.: +86-024-83684491

**Abstract:** Due to the stratified nature of the manufacturing process, material extrusion (ME) parts have lower mechanical properties than those fabricated by traditional technology. This is one of the most significant defects hindering the development and application of this rapid prototyping technique. In this paper, vibration was applied to the ME process by using piezoelectric ceramics for the first time to improve the mechanical properties of the built parts. The vibrating ME equipment was established, and the specimens processed in different build directions were individually fabricated without applied vibration and with different applied vibrations. To quantify the effect of applied vibration on their mechanical properties and to summarize the influencing rule, a series of experimental tests were then performed on these specimens. A comparison between the testing results shows that the tensile strength and plasticity of the specimens, especially those processed in the Z direction, can be obviously improved by applied vibration. The orthogonal anisotropy is decreased obviously. The improvement becomes greater with increasing vibration frequency or amplitude. From the microscopic point of view, it can be seen that applied vibration can reduce the part's defects of porosity and inclusion as well as separation between layers and, thereby, improve the bonding strength.

**Keywords** material extrusion; mechanical property; piezoelectric ceramics; applied vibration; experimental test

## 1. Introduction

Material extrusion (ME), one of the most promising rapid prototyping technologies, has been increasingly used to manufacture functional parts for electronic, automotive, aerospace, and bioengineering applications [1–3]. Despite the fact that it has been studied for a few decades, the layer-by-layer manufacturing process leads to obvious defects in the parts, such as porosity and no interlayer pressure. This leads to lower mechanical properties of the built parts compared to those manufactured by traditional technology and limits their applications. Therefore, improving the mechanical properties of the built parts is one of the key development orientations of the ME technique. Fortunately, the research and development work to fine-tune it and improve the mechanical performance of the built parts has been ongoing all the time.

Rahim et al. [4–6] mixed the polymer raw material with materials of high mechanical strength (glass fiber, ceramics, carbon fiber, etc.). The test results showed that the

mechanical and thermodynamic properties of ME parts could be effectively improved. However, this method changes the original material, and it is complicated and costly. A large number of scholars [7–13] have found that optimizing ME processing parameters (i.e., extrusion width, layer height, build direction, processing speed, nozzle diameter, infill condition, number of contours, extruder temperature, raster angle and gap, etc.) could help enhance the built products' mechanical properties. However, the optimized processing parameters are not universal (only applicable for the specified equipment), and the improvement on the built parts' forming quality is limited as the allowance for these parameter values is small. Narahara [14] used the atmospheric pressure plasma to enhance the hydrophilicity between ME processing layers. It was shown that the inter-layer bonding strength of the built parts can be effectively improved (nearly up by 15%) and, thus, the parts' mechanical properties were enhanced. However, this method decreases the surface quality of the parts and consumes too much energy. A system able to automatically detect weak links in the CAD model was developed by Stava [15]. The revised parts' strength could be obviously increased by local thickening, support increase, and weight reduction. However, this method dramatically alters the original design's structure and appearance. Li et al. [16,17] carried out the post-processing treatment of ultrasonic vibration on ME specimens to improve their mechanical properties. The results showed that the tensile strength and Young's modulus of the processed parts were significantly increased and the surface quality was enhanced. In the field of rapid prototyping, Foroozmehr [18] pioneered introducing vibration (onto the platform) into the manufacturing process. Vibrating laser powder deposition (LPD) equipment was set up mainly by combining a five-axis CNC machining center, a material powder delivery system, a high-power laser system, and an electromagnetic exciter. The exciter, connected to the platform, was to introduce and control the vibration (amplitude, frequency, and direction). The test results showed that applying vibration could effectively reduce the defect of pores in the LPD parts by 80% and thereby improve the parts' mechanical strength and ductility. Jiang [19,20] introduced vibration into the ME process by using two vibrating motors fixed on the extrusion liquefier and performed a series of experimental test, as well as theoretical and computational analyses of the melt flow behavior of polylactic acid (PLA) in the extrusion liquefier. The results show that the applied vibration can significantly improve the forming quality of the built parts. However, there is no information available about the influencing rule of different types of vibration on the ME parts' tensile property. The corresponding mechanism needs to be investigated accordingly.

In this paper, piezoelectric ceramics were combined with ME equipment to introduce different types of vibrations into the forming process. The way the different types of applied vibrations (different frequency or amplitude) worked in the internal structure and influenced the final performance of the component was revealed. Section 2 introduces experimental analysis, including the establishment of vibrating ME equipment, specimen preparation, tensile test, and scanning electron microscopy. Corresponding results and discussions are summarized in Section 3, followed by the conclusions.

## 2. Experimental Analysis

To study the mechanical properties of ME plates, this section introduces vibrating ME equipment establishment, specimen preparation, as well as related experimental analysis.

### 2.1. Vibrating ME Equipment

The ME rapid prototyping equipment (D-force V2, as shown in Figure 1) was used to fabricate the specimens for the experiment. To apply vibration to the ME process, the piezoelectric ceramic (P-5 I type, size $40 \times 10 \times 0.3$ mm$^3$) was fixed on the extrusion liquefier and the simple harmonic vibration generated by the signal generator (model No: VC2015H) was amplified by 15 times by the amplifier (model No: HPV-3C0150A0300D)

to provide high-stability and high-resolution voltage for the piezoelectric ceramic so that the extrusion liquefier was in the longitudinal vibration field. Then the actual vibration state of the extrusion liquefier was determined by the acceleration sensor (B&K4517), data acquisition card (NI USB 4431), and other vibration picking devices. Figure 1 shows the schematic diagram of the vibrating ME equipment. The signal generator has the function of adjusting the vibration frequency and the input voltage so that the frequency and amplitude of vibration can be controlled independently.

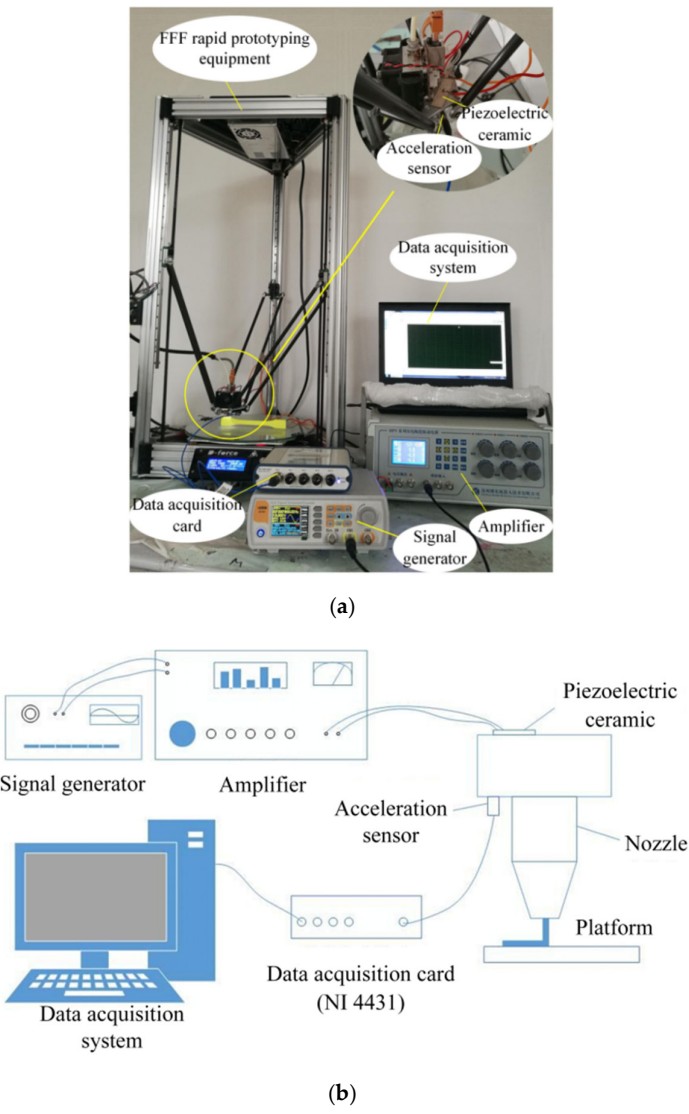

(**a**)

(**b**)

**Figure 1.** The vibrating material extrusion (ME) equipment. (**a**) Physical drawing of the vibrating ME equipment and (**b**) schematic diagram of the vibrating ME equipment.

### 2.2. Specimen Preparation

Since the components fabricated by ME are anisotropic [21,22], it is necessary to characterize their mechanical properties in different directions. To achieve this, the above vibrating ME equipment was used to prepare the tensile test specimen according to ISO 527-2-2012. Figure 2 shows the dimensions of the specimens, of which, the length is 158 mm, the testing width is 10 mm, and the thickness is 2.4 mm. The specimen material is polylactic acid (Zhuhai Tianwei Feima Printing Consumables Co. Ltd, Zhuhai city,

China), a biodegradable and renewable material with good thermal stability and strong biocompatibility. Therefore, it has been widely used in the field of additive manufacturing and plastic processing [23].

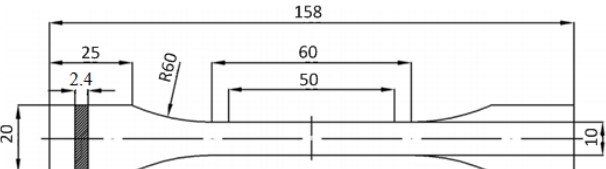

**Figure 2.** Tensile test specimen.

A total of 216 specimens were processed in this study, and they are divided into two kinds: 108 of them were built in the longitudinal direction (or X direction, parallel to the stretching direction); another 108 specimens were in the transverse direction (or Z direction, perpendicular to the stretching direction). Each kind of specimen was further divided into three different types. The 1st type included the ordinary specimens, which were represented by $X_0^0\_i$ and $Z_0^0\_i$ ($i = 1$–9); the 2nd type included the specimens built in a 0.1 g (acceleration) vibration field with different vibration frequencies, which were, respectively, 100, 200, 300, 400, 500, 600, 700, 800, and 900 Hz. These specimens were represented by $X_{100}^{0.1}\_i$, $X_{200}^{0.1}\_i$, $X_{300}^{0.1}\_i$, $X_{400}^{0.1}\_i$, $X_{500}^{0.1}\_i$, $X_{600}^{0.1}\_i$, $X_{700}^{0.1}\_i$, $X_{800}^{0.1}\_i$, $X_{900}^{0.1}\_i$, $Z_{100}^{0.1}\_i$, $Z_{200}^{0.1}\_i$, $Z_{300}^{0.1}\_i$, $Z_{400}^{0.1}\_i$, $Z_{500}^{0.1}\_i$, $Z_{600}^{0.1}\_i$, $Z_{700}^{0.1}\_i$, $Z_{800}^{0.1}\_i$, and $Z_{900}^{0.1}\_i$ ($i = 1$–9). The 3rd type included the specimens built in a 700 Hz vibration field with different vibration amplitudes, which were individually 0.1, 0.2, and 0.3 g (acceleration). They were represented by $X_{700}^{0.1}\_i$, $X_{700}^{0.2}\_i$, $X_{700}^{0.3}\_i$, $Z_{700}^{0.1}\_i$, $Z_{700}^{0.2}\_i$, and $Z_{700}^{0.3}\_i$ ($i = 1$–9); except for the frequency or amplitude of the applied vibration, all the other processing parameters were the same, such as layer height (0.15 mm), extrusion width (0.4 mm), printing speed (60 mm/s), and extrusion temperature (200 °C), etc. Table 1 shows the details.

**Table 1.** Detailed design of the specimens.

| Specimen ($i = 1$–9) | Build Direction | Vibration Frequency (Hz) | Vibration Amplitude (g) | Extrusion Width (mm) | Printing Speed (mm/s) | Extruder Temperature (°C) |
|---|---|---|---|---|---|---|
| $X_0^0\_i$ | | 0 | 0 | | | |
| $X_{100}^{0.1}\_i$ | | 100 | 0.1 | | | |
| $X_{200}^{0.1}\_i$ | | 200 | 0.1 | | | |
| $X_{300}^{0.1}\_i$ | | 300 | 0.1 | | | |
| $X_{400}^{0.1}\_i$ | | 400 | 0.1 | | | |
| $X_{500}^{0.1}\_i$ | X direction | 500 | 0.1 | 0.4 | 60 | 200 |
| $X_{600}^{0.1}\_i$ | | 600 | 0.1 | | | |
| $X_{700}^{0.1}\_i$ | | | 0.1 | | | |
| $X_{700}^{0.2}\_i$ | | 700 | 0.2 | | | |
| $X_{700}^{0.3}\_i$ | | | 0.3 | | | |
| $X_{800}^{0.1}\_i$ | | 800 | 0.1 | | | |
| $X_{900}^{0.1}\_i$ | | 900 | 0.1 | | | |
| $Z_0^0\_i$ | | 0 | 0 | | | |
| $Z_{100}^{0.1}\_i$ | Z direction | 100 | 0.1 | 0.4 | 60 | 200 |
| $Z_{200}^{0.1}\_i$ | | 200 | 0.1 | | | |
| $Z_{300}^{0.1}\_i$ | | 300 | 0.1 | | | |

| | | | |
|---|---|---|---|
| $Z_{400}^{0.1}\_i$ | | 400 | 0.1 |
| $Z_{500}^{0.1}\_i$ | | 500 | 0.1 |
| $Z_{600}^{0.1}\_i$ | | 600 | 0.1 |
| $Z_{700}^{0.1}\_i$ | | | 0.1 |
| $Z_{700}^{0.2}\_i$ | | 700 | 0.2 |
| $Z_{700}^{0.3}\_i$ | | | 0.3 |
| $Z_{800}^{0.1}\_i$ | | 800 | 0.1 |
| $Z_{900}^{0.1}\_i$ | | 900 | 0.1 |

### 2.3. Tensile Test

To determine the tensile properties of the specimens, the testing machine Shimadzu EHF-EV200k2-040 was used to perform tensile test on the 216 specimens according to the ISO 527-2-2012 standard, as shown in Figure 3. During the test, the loading rate was 5 mm/min, slow enough to obtain stable results. Since the mechanical strength of PLA is much lower than that of steel, the clamping force at both ends of each specimen during the test was set to 5 MPa so as to avoid breaking them and ensure measurement accuracy [23].

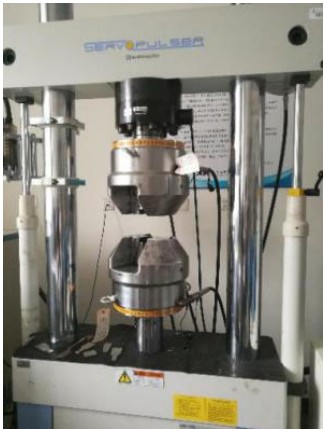 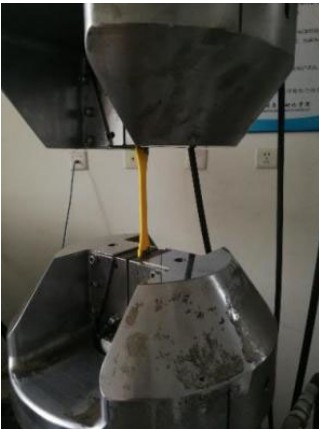

**Figure 3.** The servo-hydraulic machine for tensile test and the specimen.

### 2.4. Scanning Electron Microscopy Test

To further evaluate the forming quality of ME parts, scanning electron microscope (SEM) tests were performed with the field-emission SEM (Model: Zeiss ULTRA 55) to microscopically identify the parts' defects, mainly inclusion and pores. This can help understand the built parts' reliability and failure mechanism. For each specimen, multiple random positions were measured to ensure the representativeness and repeatability of the results. The testing steps are mainly (1) preparing samples (cutting each sample flat on one side, cleaning it, and then performing spray gold and vacuum treatment); (2) placing samples properly; (3) adjusting the observation focal length; and (4) observing and saving the data.

## 3. Results and Discussion

The effect of different applied vibrations (different frequencies or amplitudes) on the specimen's mechanical properties is detailed in this section. The influencing rule is summarized accordingly.

### 3.1. Effect of Different Frequencies

#### 3.1.1. Z-Direction Specimens

The stress-strain relationship between specimens $Z_0^0\_i$, $Z_{100}^{0.1}\_i$, $Z_{200}^{0.1}\_i$, $Z_{300}^{0.1}\_i$, $Z_{400}^{0.1}\_i$, $Z_{500}^{0.1}\_i$, $Z_{600}^{0.1}\_i$, $Z_{700}^{0.1}\_i$, $Z_{800}^{0.1}\_i$, and $Z_{900}^{0.1}\_i$ ($i$ = 1–9) was compared and analyzed in Figure 4. The influence of applied vibrations of the same acceleration amplitude (0.1 g) but different frequencies (0–900 Hz) on the tensile properties of the specimens can be determined. As can be seen, the tensile properties (tensile strength and plasticity) of the specimens built with applied vibration have been greatly improved, and they are further enhanced with the increase in frequency. Table 2 shows the details.

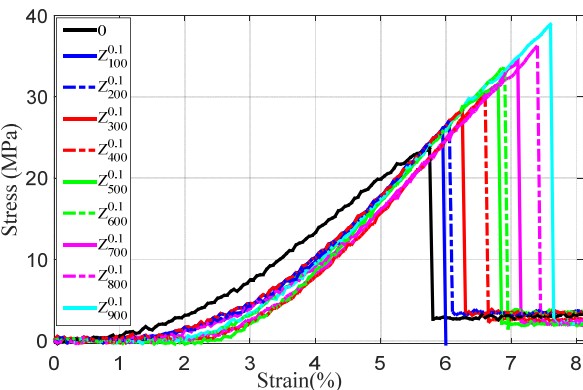

**Figure 4.** The effect of applied vibration (0.1 g amplitude with different frequencies) on the tensile properties of Z-direction specimens.

**Table 2.** The tensile properties of the Z-direction specimens processed without applied vibration and with different applied vibrations.

| Specimens ($i$ = 1–9) | $Z_0^0\_i$ | $Z_{100}^{0.1}\_i$ | $Z_{200}^{0.1}\_i$ | $Z_{300}^{0.1}\_i$ | $Z_{400}^{0.1}\_i$ | $Z_{500}^{0.1}\_i$ | $Z_{600}^{0.1}\_i$ | $Z_{700}^{0.1}\_i$ | $Z_{800}^{0.1}\_i$ | $Z_{900}^{0.1}\_i$ |
|---|---|---|---|---|---|---|---|---|---|---|
| Average tensile strength (MPa) | 23.68 | 26.16 | 27.06 | 28.31 | 30.35 | 31.59 | 33.44 | 34.65 | 36.27 | 38.92 |
| Standard deviation | 1.61 | 1.34 | 1.29 | 1.22 | 1.16 | 1.02 | 0.92 | 0.85 | 0.73 | 0.71 |
| Growth (%) | - | 10.5 | 14.3 | 17.3 | 28.2 | 33.4 | 41.2 | 46.3 | 53.2 | 64.3 |
| Average plasticity (%) | 5.7 | 5.95 | 6.1 | 6.3 | 6.65 | 6.85 | 6.95 | 7.14 | 7.4 | 7.65 |
| Standard deviation | 0.36 | 0.30 | 0.28 | 0.26 | 0.26 | 0.24 | 0.22 | 0.18 | 0.16 | 0.14 |
| Growth (%) | - | 4.4 | 7.0 | 10.5 | 16.7 | 20.2 | 21.9 | 25.3 | 29.8 | 34.2 |

It can be seen that the average tensile strength of the ordinary specimen (without applied vibration) $Z_0^0\_i$ is 23.68 MPa and the average plasticity (the ability to resist deformation under a certain external force, expressed by elongation here) is 5.7%; when vibration is introduced, both specimens' tensile strength and plasticity are obviously increased. For example, the average tensile strength and plasticity of specimen $Z_{100}^{0.1}\_i$ are 26.16 MPa and 5.95%, respectively, and they are increased by 10.5% and 4.4%, respectively, compared with those of $Z_0^0\_i$. In addition, the average values for specimen $Z_{900}^{0.1}\_i$ are, respectively, 38.92 MPa and 7.65%, with the growth being 64.3% and 34.2%, respectively. This is because the layer in the Z direction is deposited above the other and it is perpendicular to the applied load during the tensile test. The Z-direction specimens'

tensile properties mainly rely on the bonding strength between material fibers instead of the fibers themselves. Applied vibration oscillates the molten pool inside the extrusion liquefier, making the gas escape quickly and reducing the occurrence of pores and thermal cracks in the extrudate. Meanwhile, the melt apparent viscosity is decreased and its fluidity is enhanced, making the extrudate finer and more uniform [20]. In addition, the applied vibration generates a reciprocating excitation pressure in the ME process, assisting to make the connection between the layers closer. With an increase in the applied vibration frequency, these improvements will be further enhanced, leading to a better forming quality. Therefore, the tensile performance of Z-direction specimens built with applied vibration is better than that of the normal ones, and the property will be further improved with increasing vibration frequency.

Figure 5 shows the SEM micrographs of the specimens' cross sections. As can be seen, there are fewer layer separations in specimens to which vibrations were applied, and the bonding gap is much smaller. The layer height and extrusion width are more uniform, with less distortion and deformation. With increasing vibration frequency, the defects mentioned above are further reduced. Therefore, applied vibration can significantly improve the forming quality of ME parts and it will be further improved with increasing vibration frequency.

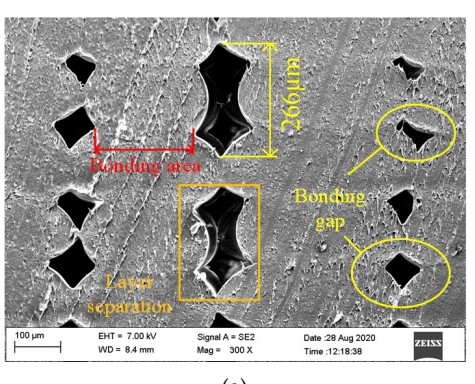

(**a**)

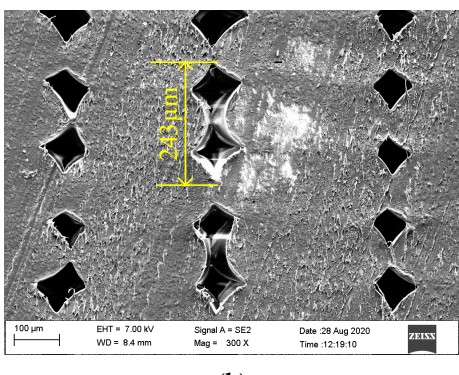

(**b**)

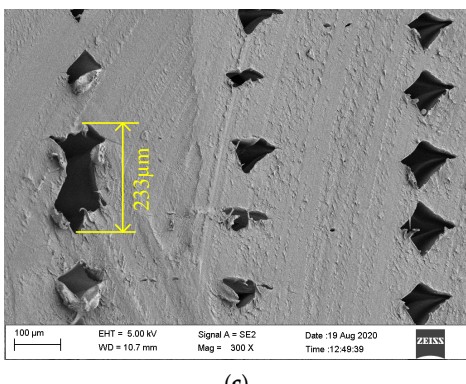

(**c**)

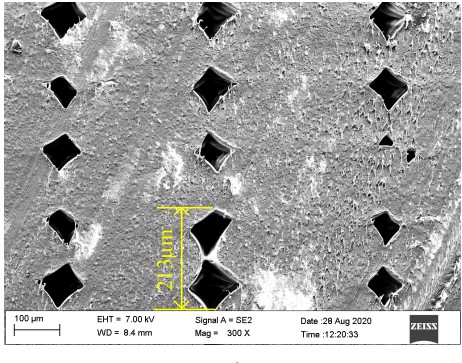

(**d**)

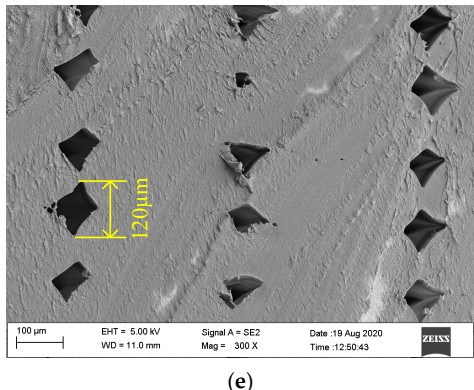 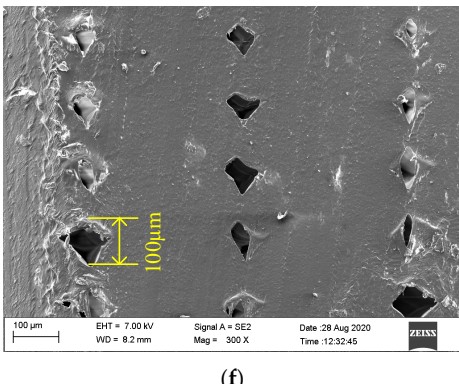

(**e**)　　　　　　　　　　　　　　　　　　　　　　（**f**）

**Figure 5.** SEM micrographs of specimens' cross sections. (**a**) Without applied vibration; (**b**) effect of 100 Hz and 0.1 g vibration applied; (**c**) effect of 300 Hz and 0.1 g vibration applied; (**d**) effect of 500 Hz and 0.1 g vibration applied; (**e**) effect of 700 Hz and 0.1 g vibration applied; and (**f**) effect of 900 Hz and 0.1 g vibration applied.

### 3.1.2. X-Direction Specimens

Similarly, the stress-strain relationship of $X_0^0\_i$, $X_{100}^{0.1}\_i$, $X_{200}^{0.1}\_i$, $X_{300}^{0.1}\_i$, $X_{400}^{0.1}\_i$, $X_{500}^{0.1}\_i$, $X_{600}^{0.1}\_i$, $X_{700}^{0.1}\_i$, $X_{800}^{0.1}\_i$, and $X_{900}^{0.1}\_i$ (*i* = 1–9) is compared in Figure 6. The influence of the applied vibrations of the same amplitude (0.1 g) but different frequencies (0–900 Hz) on the tensile properties of the X-direction specimens can be determined. It is shown that the tensile properties (tensile strength and plasticity) of the specimens have only been slightly changed with applied vibration, and they are almost unchanged with the increase in frequency. Table 3 shows the details.

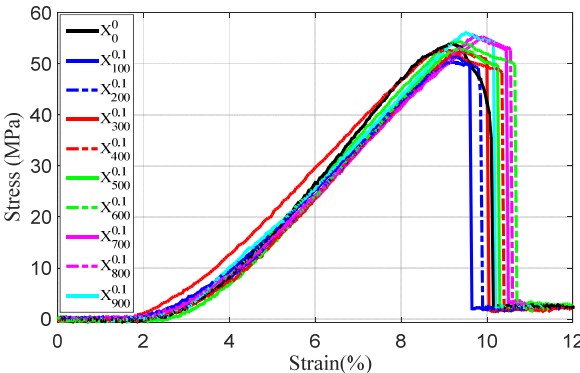

**Figure 6.** The effect of applied vibration (0.1 g amplitude with different frequencies) on the tensile properties of X-direction specimens.

**Table 3.** The tensile properties of the X-direction specimens processed without applied vibration and with different applied vibrations.

| Specimens ($i$ = 1–9) | $X_0^0\_i$ | $X_{100}^{0.1}\_i$ | $X_{200}^{0.1}\_i$ | $X_{300}^{0.1}\_i$ | $X_{400}^{0.1}\_i$ | $X_{500}^{0.1}\_i$ | $X_{600}^{0.1}\_i$ | $X_{700}^{0.1}\_i$ | $X_{800}^{0.1}\_i$ | $X_{900}^{0.1}\_i$ |
|---|---|---|---|---|---|---|---|---|---|---|
| Average tensile strength (MPa) | 53.85 | 50.08 | 51.47 | 52.36 | 52.58 | 52.93 | 54.36 | 55.45 | 55.22 | 55.99 |
| Standard deviation | 1.99 | 1.72 | 1.63 | 1.86 | 1.62 | 1.83 | 1.58 | 1.92 | 1.84 | 1.76 |
| Growth (%) | - | −7 | −4.42 | −2.85 | −2.36 | −1.71 | 0.95 | 2.97 | 2.54 | 3.97 |
| Average plasticity (%) | 10.1 | 9.65 | 9.8 | 10 | 10.4 | 10.3 | 10.7 | 10.5 | 10.6 | 10.2 |
| Standard deviation | 0.53 | 0.51 | 0.38 | 0.33 | 0.41 | 0.36 | 0.47 | 0.43 | 0.50 | 0.39 |
| Growth (%) | - | −4.46 | −3.06 | −0.99 | 10.6 | 2.97 | 5.94 | 3.96 | 4.95 | 0.99 |

As can be seen, the average tensile strength of $X_0^0\_i$ is 53.85 MPa and the average plasticity is 10.1%; when vibration is introduced, the tensile properties are only slightly changed. For example, the average tensile strength and plasticity of $X_{100}^{0.1}\_i$ are, respectively, 50.08 MPa and 9.65%, and they are changed by −7% and −4.46%, respectively. In addition, these values of $X_{900}^{0.1}\_i$ are, respectively, 55.99 MPa and 10.2%, with the growth being 3.97% and 0.99%, respectively. The main reason is that the X-direction specimens' fiber layers are parallel to the applied load during the test and the tensile properties are mainly dependent on the strength of the material fibers themselves. Although applied vibration can significantly improve the specimens' forming quality, it has little effect on the fibers' property. Therefore, the applied vibrations' influence on the tensile properties of X-direction specimens is limited, which can be ignored [19].

### 3.2. Effect of Different Amplitudes

3.2.1. Z-Direction Specimens

According to the same process, the stress-strain relationship of $Z_0^0\_i$, $Z_{700}^{0.1}\_i$, $Z_{700}^{0.2}\_i$, and $Z_{700}^{0.3}\_i$ ($i$ = 1–9) is compared and analyzed in Figure 7. The influence of the vibration field with the same frequency (700 Hz) but different amplitudes (0.1–0.3 g) on the tensile properties of Z-direction specimens can be determined. Details are shown in Table 4.

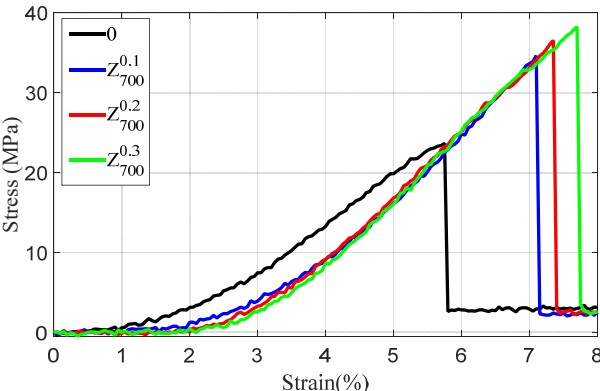

**Figure 7.** The effect of applied vibration (700 Hz frequency with different amplitudes) on the tensile properties of Z-direction specimens.

**Table 4.** The tensile properties of the Z-direction specimens processed without applied vibration and with different applied vibrations.

| Specimens ($i$ = 1–9) | $Z_0^0\_i$ | $Z_{700}^{0.1}\_i$ | $Z_{700}^{0.2}\_i$ | $Z_{700}^{0.3}\_i$ |
|---|---|---|---|---|
| Average tensile strength (MPa) | 23.68 | 34.65 | 36.54 | 38.25 |
| Standard deviation | 1.61 | 0.85 | 0.82 | 0.78 |
| Growth (%) | - | 46.3 | 54.3 | 61.5 |
| Average plasticity (%) | 5.7 | 7.14 | 7.35 | 7.7 |
| Standard deviation | 0.36 | 0.18 | 0.17 | 0.14 |
| Growth (%) | - | 25.3 | 28.9 | 35.1 |

The average tensile strength of $Z_{700}^{0.1}\_i$, $Z_{700}^{0.2}\_i$, and $Z_{700}^{0.3}\_i$ are, respectively, 34.65, 36.54, and 38.25 MPa, and they are, correspondingly, increased by 46.3, 54.3, and 61.5% compared with $Z_0^0\_i$. In terms of plasticity, the corresponding values are 7.14, 7.35, and 7.7%, increased by 25.3, 28.9, and 35.1%, respectively. Therefore, increasing the amplitude of the applied vibration is also able to further improve the tensile properties of ME specimens built in the Z direction. This is due to a similar reason as explained in Section 3.1.1. Figure 8 shows the SEM micrographs of the specimens' cross sections. It shows that with the increase in the vibration amplitude, the bonding area is increased and the bonding gap becomes smaller. There are fewer defects of layer separation, distortion, and deformation.

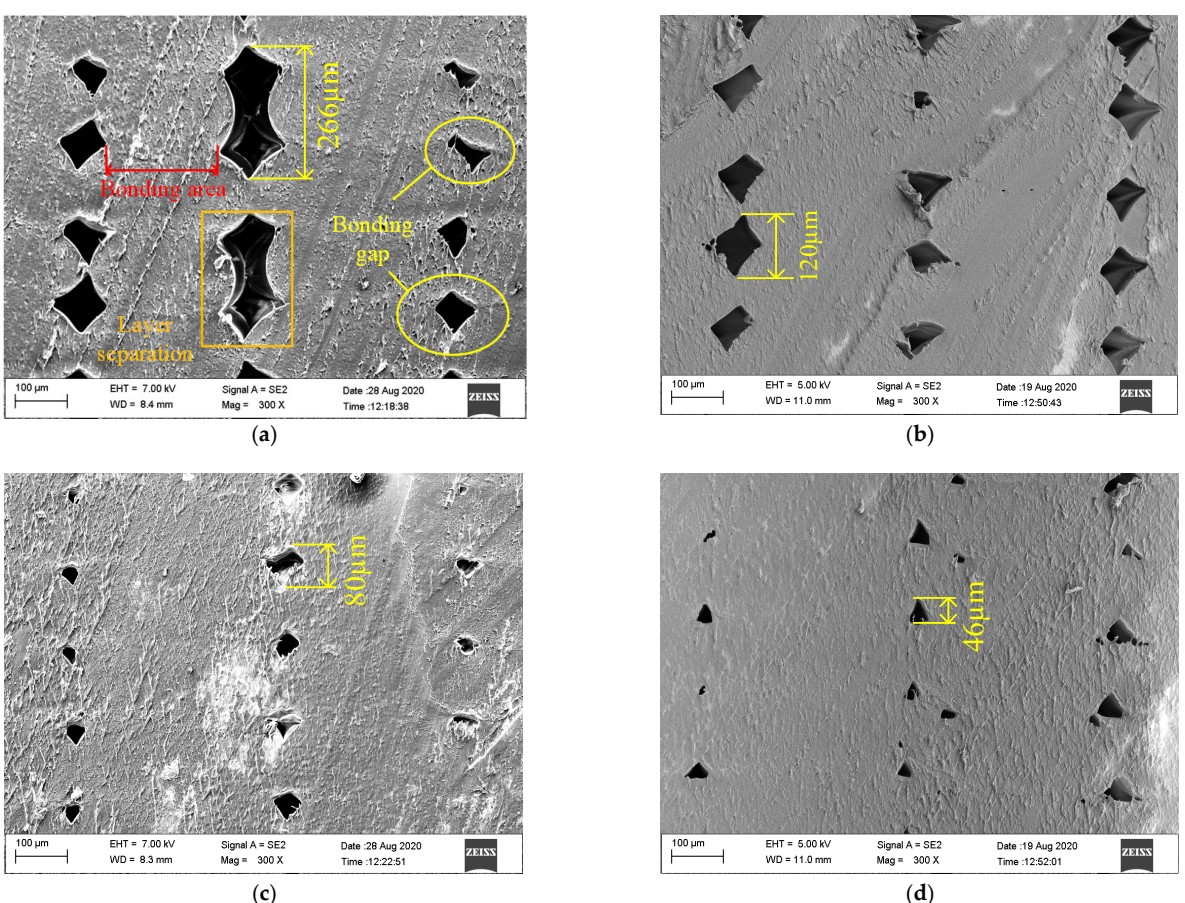

**Figure 8.** SEM micrographs of specimens' cross sections. (**a**) Without applied vibration; (**b**) effect of 700 Hz and 0.1 g vibration applied; (**c**) effect of 700 Hz and 0.2 g vibration applied; and (**d**) effect of 700 Hz and 0.3 g vibration applied.

### 3.2.2. X-Direction Specimens

Figure 9 compares and analyzes the stress-strain relationship of $X_0^0\_i$, $X_{700}^{0.1}\_i$, $X_{700}^{0.2}\_i$, and $X_{700}^{0.3}\_i$ (i = 1–9). The influence of the vibration field with the same frequency (700 Hz) but different amplitudes (0.1–0.3 g) on the tensile properties of the X-direction specimens can be determined. It can be seen that there is almost no change in the specimens' tensile properties with increasing amplitude. Details are shown in Table 5. When vibration is introduced, the tensile properties of the X-direction specimens are only slightly changed. The average tensile strength of $X_{700}^{0.1}\_i$, $X_{700}^{0.2}\_i$, and $X_{700}^{0.3}\_i$ are, respectively, 55.45, 55.13, and 54.32 MPa, increasing only by 2.97, 2.38, and 0.87%, respectively, compared with $X_0^0\_i$. In terms of plasticity, the corresponding values are 10.9, 10.8, and 11.0%, increasing by 7.92, 6.93, and 8.91% respectively. This can be explained as proposed in Section 3.1.2.

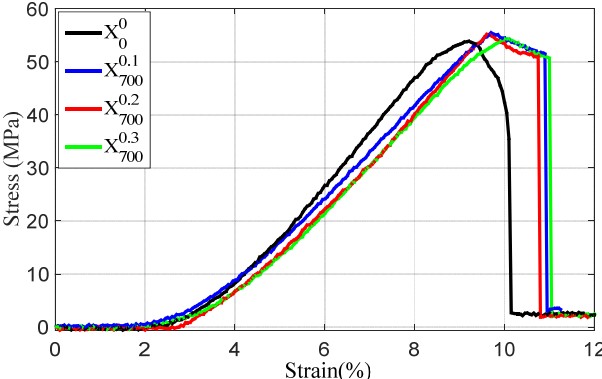

**Figure 9.** The effect of applied vibration (700 Hz frequency with different amplitudes) on the tensile properties of X-direction specimens.

**Table 5.** The tensile properties of the X-direction specimens processed without applied vibration and with different applied vibrations.

| Specimens (*i* = 1–9) | $X_0^0\_i$ | $X_{700}^{0.1}\_i$ | $X_{700}^{0.2}\_i$ | $X_{700}^{0.3}\_i$ |
|---|---|---|---|---|
| Average tensile strength (MPa) | 53.85 | 55.45 | 55.13 | 54.32 |
| Standard deviation | 1.99 | 1.92 | 1.86 | 1.88 |
| Growth (%) | — | 2.97 | 2.38 | 0.87 |
| Average plasticity (%) | 10.1 | 10.9 | 10.8 | 11.0 |
| Standard deviation | 0.53 | 0.43 | 0.46 | 0.40 |
| Growth (%) | — | 7.92 | 6.93 | 8.91 |

### 3.3. Anisotropy of Tensile Properties

### 3.3.1. Effect of Different Frequencies of Applied Vibration

Figure 10 compares the tensile properties of the Z- and X-direction specimens processed without applied vibration and with the application of vibrations of different frequencies. It is shown that applied vibration can significantly reduce the tensile property anisotropy of ME parts and it can further decrease the anisotropy with an increase in its frequency. Details can be found in Table 6.

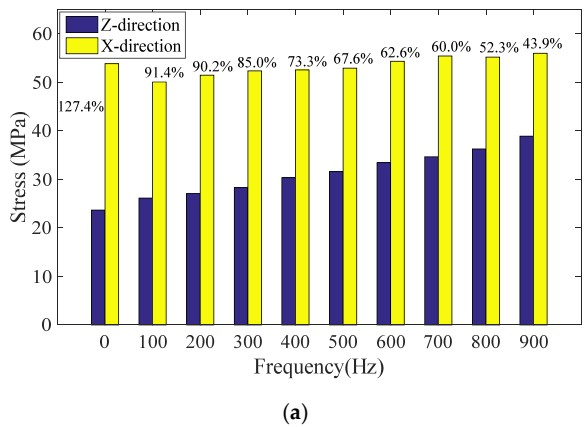 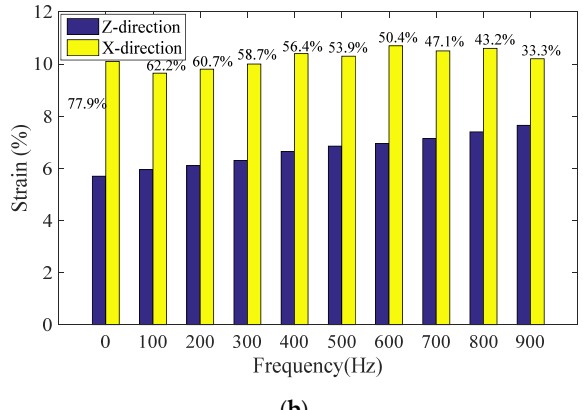

(**a**)                                    (**b**)

**Figure 10.** Tensile property anisotropy of the specimens built without applied vibration and with different vibrations applied. (**a**) The effect of different vibrations (0.1 g amplitude with different frequencies) on the specimens' tensile strength anisotropy and (**b**) the effect of different vibrations (0.1 g amplitude with different frequencies) on the specimens' plasticity anisotropy.

**Table 6.** The tensile property anisotropy of the specimens built without applied vibration and with vibrations of different frequencies applied.

| Specimens ($i$ = 1–9) | $Z_0^0\_i$ $X_0^0\_i$ | $Z_{100}^{0.1}\_i$ $X_{100}^{0.1}\_i$ | $Z_{200}^{0.1}\_i$ $X_{200}^{0.1}\_i$ | $Z_{300}^{0.1}\_i$ $X_{300}^{0.1}\_i$ | $Z_{400}^{0.1}\_i$ $X_{400}^{0.1}\_i$ | $Z_{500}^{0.1}\_i$ $X_{500}^{0.1}\_i$ | $Z_{600}^{0.1}\_i$ $X_{600}^{0.1}\_i$ | $Z_{700}^{0.1}\_i$ $X_{700}^{0.1}\_i$ | $Z_{800}^{0.1}\_i$ $X_{800}^{0.1}\_i$ | $Z_{900}^{0.1}\_i$ $X_{900}^{0.1}\_i$ |
|---|---|---|---|---|---|---|---|---|---|---|
| Average tensile strength difference (%) | 127.4 | 91.4 | 90.2 | 85.0 | 73.3 | 67.6 | 62.6 | 60.0 | 52.3 | 43.9 |
| Average plasticity difference (%) | 77.9 | 62.2 | 60.7 | 58.7 | 56.4 | 53.9 | 50.4 | 47.1 | 43.2 | 33.3 |

As can be seen, the average tensile strength difference between $Z_0^0\_i$ and $X_0^0\_i$ ($i$ = 1–9) is 127.4%. The corresponding plasticity difference is 77.9%. For $X_{100}^{0.1}\_i$ and $Z_{100}^{0.1}\_i$ ($i$ = 1–9), the difference in the two parameters is reduced to 91.4% and 62.2%, respectively. When considering $X_{900}^{0.1}\_i$ and $Z_{900}^{0.1}\_i$ ($i$ = 1–9), the difference between the two values is further reduced to 43.9% and 33.3%, respectively. The reason is that when vibrations are introduced, the specimens have fewer defects and the bond between material layers becomes closer, making such specimens denser in structure with stronger adhesive strength. With increase in the vibration frequency, the forming quality of specimens is further improved, especially for the Z-direction ones. Therefore, the tensile property orthogonal anisotropy of the specimen is reduced with applied vibration and further decreased with increasing vibration frequency.

### 3.3.2. Effect of Different Amplitudes of Applied Vibration

Similarly, the effect of applied vibrations of different amplitudes on the tensile properties of Z- and X-direction specimens is compared in Figure 11. It shows that the tensile property anisotropy of the ME parts is obviously reduced, and it is further decreased with increasing amplitude. Table 7 lists the details.

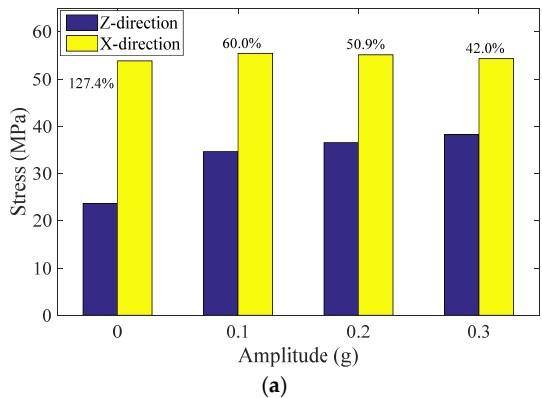
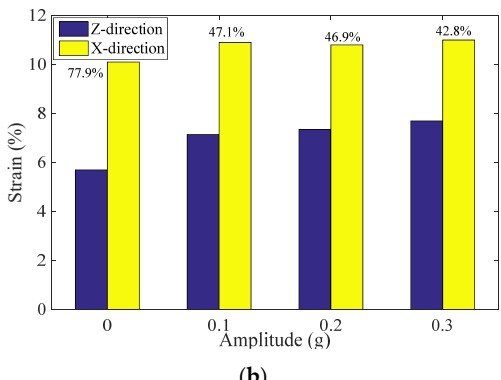

(**a**)  (**b**)

**Figure 11.** Tensile property anisotropy of the specimens processed without applied vibration and with different applied vibrations. (**a**) The effect of different vibrations (700 Hz frequency with different amplitudes) on the specimens' tensile strength anisotropy and (**b**) the effect of different vibrations (700 Hz frequency with different amplitudes) on the specimens' plasticity anisotropy.

**Table 7.** The tensile property anisotropy of the specimens processed without applied vibration and with vibrations of different amplitudes.

| Specimens ($i$ = 1–9) | $Z_0^0\_i$ $X_0^0\_i$ | $Z_{700}^{0.1}\_i$ $X_{700}^{0.1}\_i$ | $Z_{700}^{0.2}\_i$ $X_{700}^{0.2}\_i$ | $Z_{700}^{0.3}\_i$ $X_{700}^{0.3}\_i$ |
|---|---|---|---|---|
| Average tensile strength difference (%) | 127.4 | 60.0 | 50.9 | 42.0 |
| Average plasticity difference (%) | 77.9 | 47.1 | 46.9 | 42.8 |

As can be seen, the difference in the average tensile strength and plasticity between $X_{700}^{0.1}\_i$ and $Z_{700}^{0.1}\_i$ ($i$ = 1–9) is, respectively, reduced to 60.0% and 47.1% compared with those of the ordinary specimens ($Z_0^0\_i$ and $X_0^0\_i$). While for $X_{700}^{0.3}\_i$ and $Z_{700}^{0.3}\_i$ ($i$ = 1–9), the difference in the two values is further reduced to 42.0% and 42.8%, respectively. The reason is similar to the one mentioned above.

## 4. Conclusions

In this paper, the influence of different applied vibrations on the tensile properties (tensile strength and plasticity) of ME parts is studied experimentally. Specific conclusions are as follows:

(1) Applying vibration during the ME process can obviously improve the tensile strength and plasticity of Z-direction specimens and further enhance them with an increase in the vibration frequency or the amplitude. However, the effect on the specimens built in the X direction is small to negligible.

(2) Applied vibration can greatly reduce the anisotropy of the ME parts, which can be further reduced with an increase in the vibration frequency or the amplitude.

(3) The SEM analysis confirms that the specimens processed with applied vibration have fewer defects and better forming quality than the ordinary ones. With increasing vibration frequency or amplitude, the specimens' defects are further reduced and the forming quality is further improved.

(4) The proposed novel method, introducing vibration into the ME process by using piezoelectric ceramics, could make such specimens denser in structure and with stronger adhesive strength, thereby improve the forming quality. This method is also applicable for other additive manufacturing techniques to improve the forming quality of built parts.

**Author Contributions:** Conceptualization, S.J.; data curation, T.D. and Y.Z.; formal analysis, S.J., T.D., and Y.Z.; funding acquisition, S.J.; investigation, T.D., Y.Z., and W.D.; methodology, S.J. and M.Z.; project administration, S.J.; resources, S.J., T.D., and M.Z.; software, T.D.; supervision, S.J.; validation, S.J., T.D., and Y.Z.; writing—original draft preparation, T.D. and Y.Z.; writing—review and editing, S.J. and W.D. All authors have read and agreed to the published version of the manuscript.

**Funding:** The work described was supported by the National Natural Science Foundation of China (51705068) and the fundamental research funds for central universities (N180703009 and N170302001).

**Informed Consent Statement:** I confirm that the manuscript has been submitted solely to the journal and that it is not published, in press, or submitted elsewhere. I confirm that all the research meets the ethical guidelines, including adherence to the legal requirements of the study country. I confirm that I have seen, read, and understood the guidelines on copyright. I confirm that the names of all the co-authors have been included in the manuscript and that these co-authors all had an active part in the final manuscript.

**Data Availability Statement:** All available. Contact the corresponding author for data.

**Conflicts of Interest:** No potential conflict of interest is reported by the authors.

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
