# Peer review of "Experimental Study on Improving the Mechanical Properties of Material Extrusion Rapid Prototyping Polylactic Acid Parts by Applied Vibration"

_applsci, doi:10.3390/app11041820_

Round 1
Reviewer 1 Report
Review of: Experimental study on improving the mechanical property of material extrusion rapid prototyping parts by applied vibration
In this study, a vibration technique is introduced in order to reduce porosities in material extrusion additive manufacturing and thereby improve the mechanical properties. The paper is well written and structured, and the theme is of interested for the readers of the journal. I have the following major issues that needs to be addressed before I can recommend an acceptance of the paper.
Major:
- The vibration device is mounted on the print head it seems, cf. Fig. 1. This means that the vibration improves the flowability of the material that is exiting the print head. However, why not just improve the flowability by increasing the printing temperature?
- Having a vibration devise on the print head must affect the geometrical precision of the print. This geometrical deviation must be reported, so the reader can assess whether the loss of geometrical accuracy can be justified by the improved mechanical properties.
- You need to quantify the porosities in the SEM pictures, so it is easier for the reader to see the improvement.
Minor:
Line 84: so that the molten nozzle is in the longitudinal vibration field – what do you mean with ‘molten nozzle’?
Line 106: You need a punctuation before ‘They’.
Author Response
Dear Reviewer,
Thanks a lot for your feedback. The following shows the author’s response to your comments and suggestions. The revised content is highlighted in cyan color in the manuscript.
Reviewers 1:
In this study, a vibration technique is introduced in order to reduce porosities in material extrusion additive manufacturing and thereby improve the mechanical properties. The paper is well written and structured, and the theme is of interested for the readers of the journal. I have the following major issues that needs to be addressed before I can recommend an acceptance of the paper.
Detailed comments:
- The vibration device is mounted on the print head it seems, cf. Fig. 1. This means that the vibration improves the flowability of the material that is exiting the print head. However, why not just improve the flowability by increasing the printing temperature?
Response: Thanks for the good question.
Increasing the printing temperature does improve the flowability of the material, but meanwhile it can easily cause several problems, like forming difficulty, out of shape, forming precision, etc. There is a balance (or optimization) between the printing temperature and cooling, as well as the other processing parameters. Utilizing vibration during fabrication can not only increase the flowability of the material, but also have some other positive effect, such as less porosities, smaller layer spacing and bonding gap bewteen the adjacent extruded filament (as shown in Figure 6 and 9), less deformation and distortion and so on. Therefore, the method, introducing vibration into the material extrusion (ME) processing, has many positive aspects rather than just improving the material flowability, and thereby improving the forming quality of the built part. The corresponding mechanism for each aspect needs to be investigated. This manuscript focuses on the tensile property.
- Having a vibration devise on the print head must affect the geometrical precision of the print. This geometrical deviation must be reported, so the reader can assess whether the loss of geometrical accuracy can be justified by the improved mechanical properties.
Response: Thanks for the good suggestion.
The application of vibration will affect the printing accuracy, but not necessarily negative effect. Although there has been no such study up to now, the influence of applied vibration on the surface roughness of the built part was investigated by the author, and the results showed that the surface quality was improved by the applied vibration, and it was further improved with the increasing frequency or amplitude of the applied vibration, as shown in the following figures. Similarly, the effect of the applied vibration on the parts’ printing accuracy is likely positive. However, no information is available about this research aspect, and the same gose for this manuscript. It is a pretty good research objective, which the author will seriously focus on in the near future.
Fig.1 Surface roughness vertical to the fiber dicrection of the sample built with applied vibration of the same amplitude but different frequency
Fig.2 Surface roughness vertical to the fiber dicrection of the sample built with applied vibration of the same frequency but different amplitude
- You need to quantify the porosities in the SEM pictures, so it is easier for the reader to see the improvement.
Response: Thanks for the suggestion.
The author has added the dimensions of the porosities in Figure 6 and 9, please have a check.
- Line 84: so that the molten nozzle is in the longitudinal vibration field – what do you mean with ‘molten nozzle’?
Response: Sorry for the inaccurate expression.
The author has revised the whole article, and replaced "molten nozzle" with "extrusion liquefier".
- Line 106: You need a punctuation before ‘They’.
Response: Sorry for the carelessness.
The author has revised these sentences. Please have a check the highlighted content below Table 1.
The above shows the author’s response to the comments and suggestions from Reviewer 1.
Best Regards,
Shijie

Reviewer 2 Report
1) The introduction does not clarify the problem and suggested solution. The lit review is not appropriate since the manufacturing technique and more importantly, the nature of the materials used in the cited works are very different. It is not conclusive where the authors were trying to get by presenting those works! Is there the same problem in all AM techniques? Is it the same for all materials (metals, ceramics, polymers)? Are they claiming their proposed solution works for all materials systems and all AM techniques? It is not clear at all.
2) In fig.1, either present the photo or the schematic
3) Line 111, what is 0.1g vibration field? Did the authors mean the acceleration of the vibration?
4) I am already in the results and discussion section but still do not know what material are we looking at? There should be a full description of the material used for the experiments, including the composition and the application of the material. Why was it selected for this work? Lit review on the defects and mechanical properties of this material produced by other techniques! Source of the material (supplier)?
5) Line 132: " random positions were measured", what was measured using SEM? How was it measured?
6) In Line 151, the authors mentioned average plasticity is 5.7%. Can they define plasticity? Is the unit of plasticity "%" ?
7) I recommend doing some image analysis to have a quantitative analysis about the size and shape of defects in different samples.
8) Why strain was reduced in the X-direction sample vibrated at 900 Hz?
9) What are the standard deviations for each reported value in different graphs? For example, in Fig 1, panel a & b, both Z- and X-direction samples with 0.1 and 0.2 g amplitudes are almost the same! All the graphs should have the standard deviation values
10) The authors are concluding that what they proposed is universal. It is not acceptable. They did some tests on an unknown material! They did not report errors in their measurement and then concluded "Applying vibration during ME process can obviously improve the tensile strength". Also, the title of the paper implies that they are proposing a solution for all types of materials. While I did not find their work, a comprehensive research work for such a claim. To prove what they claimed they need to show results of several different families of "known" materials, including polymers, ceramics, and metals. This work can be a preliminary test for further investigation.
11) there is a lack of literature review that needs to be done for this work.
Author Response
Dear Reviewer,
Thanks a lot for your feedback. The following shows the author’s response to your comments and suggestions. The revised content is highlighted in cyan color in the manuscript.
Reviewers 2:
- The introduction does not clarify the problem and suggested solution. The lit review is not appropriate since the manufacturing technique and more importantly, the nature of the materials used in the cited works are very different. It is not conclusive where the authors were trying to get by presenting those works! Is there the same problem in all AM techniques? Is it the same for all materials (metals, ceramics, polymers)? Are they claiming their proposed solution works for all materials systems and all AM techniques? It is not clear at all.
Response: Thanks for the question and comment.
This manuscript focuses on material extrusion (ME) technology rather than all AM techniques. The literature review is all about ME technology, describing the existing methods to improve the mechanical property of the ME parts. The material used in the cited works are mainly polylactic acid (PLA) and Acrylonitrile butadiene styrene (ABS). These two types of material are widely used in ME technology. The research material in this manuscript is PLA. The author introduced it in the 1st paragraph in Section 2.2. To further clarify it, the author revised this part (see the highlighted content and added reference [23]). The author also changed the title of the article to “Experimental study on improving the mechanical property of material extrusion rapid prototyping polylactic acid parts by applied vibration”. Moreover, corresponding information is also added in Section 2.3, emphasizing the material of polylactic acid.
The literature review shows the existing methods to improve the mechanical property of the ME parts and they all have some limitations to some extent. The author proposed a novel method, that is combining piezoelectric ceramic and ME equipment to introduce vibration into the ME process. In the authors’ previous research, a series of experimental tests and theoretical and computational analysis have been performed to investigate the effect of applied vibration on the built parts’ mechanical property, dynamic characteristics and the melt flow behavior of PLA in the extrusion liquefier. The results show that the applied vibration can significantly improve the forming quality of the built parts, see the highlighted content in Section 1 (Introduction). This manuscript further investigates the influencing rule of different types of vibration (frequency or amplitude) on the tensile property of the built ME parts.
Theoretically speaking, as long as adopted in a suitable way, the method (using vibration in forming process) can be applied to the other additive manufacturing techniques and other materials according to the principle of similarity, as both ME technique and PLA material are ordinary. In Section 4 (Conclusions), the author concluded that “This method is also applicable for the other additive manufacturing techniques to improve the forming quality of the built parts.” But of course, corresponding research needs to be carried out to confirm this conclusion.
- In fig.1, either present the photo or the schematic
Response: Thanks for the suggestion.
The author wants to show the setup of the vibrating ME rapid prototyping equipment as clearly as possible. Both photo and schematic are presented here is helpful for the readers to have a clearer understanding. The author suggests keeping them both.
- Line 111, what is 0.1g vibration field? Did the authors mean the acceleration of the vibration?
Response: Thanks for the question.
Yes, 0.1g is the acceleration of the applied vibration. To clarify this, the author has added this accordingly in the paragraph below Table 1 in Section 2.2, please have a check the highlighted content.
- I am already in the results and discussion section but still do not know what material are we looking at? There should be a full description of the material used for the experiments, including the composition and the application of the material. Why was it selected for this work? Lit review on the defects and mechanical properties of this material produced by other techniques! Source of the material (supplier)?
Response: Thanks for the question and suggestion.
The author introduced the material (polylactic acid, PLA) in the 1st paragraph in Section 2.2. To further clarify it, the author revised this part according to the reviewer’s comment (see the highlighted content and added reference [23]). Moreover, the title of the manuscript has been changed, and the corresponding information is also added in Section 2.3, emphasizing the material of polylactic acid.
- Line 132: " random positions were measured", what was measured using SEM? How was it measured?
Response: Thanks for the question.
In this manucript, SEM was mainly used to microscopically observe the porosities on the cross section of the built parts at random positions or locations. The measurement procedure has been added in Section 2.4, please have a check the highlighted content.
- In Line 151, the authors mentioned average plasticity is 5.7%. Can they define plasticity? Is the unit of plasticity "%" ?
Response: Thanks for the question.
Plasticity is the ability of a solid to resist deformation under a certain external force. It is mainly expressed by elongation or shrinkage before the fracture of the solid. Therefore, the unit is % in this manuscript to show the results clearly. To clarify these, the author has added the relevant information in the 2nd line of the paragraph below Figure 5, please see the the highlighted content.
- I recommend doing some image analysis to have a quantitative analysis about the size and shape of defects in different samples.
Response: Thanks for the good suggestion.
The author has added the dimensions of the porosities in Figure 6 and 9, please have a check.
- Why strain was reduced in the X-direction sample vibrated at 900 Hz?
Response: Thanks for the question.
The fibers of X-direction specimens are parallel to the applied load during the tensile test, and the tensile properties are mainly dependent on the strength of the material fibers themselves. Although applied vibration can significantly improve the specimens’ forming quality, it has little effect on the fibers’ property. Therefore, the effect of applied vibrations on the tensile property of X-direction specimens is limited, sometimes even negative. In fact, not just at 900 Hz, we can see that at 100 and 200 Hz, the X-direction specimens’ plasticity is also reduced. The main reason for this phenomenon may be measurement error. However, the isotropy is improved with applied vibration, and it is furter improved with the increasing frequency or amplitude of the applied vibration.
- What are the standard deviations for each reported value in different graphs? For example, in Fig 1, panel a & b, both Z- and X-direction samples with 0.1 and 0.2 g amplitudes are almost the same! All the graphs should have the standard deviation values.
Response: Thanks for the question and suggestion.
To clarify this, the author has added the standard deviation values of each type of samples in Table 2, 3, 4 and 5, please have a check the highlighted content.
- The authors are concluding that what they proposed is universal. It is not acceptable. They did some tests on an unknown material! They did not report errors in their measurement and then concluded "Applying vibration during ME process can obviously improve the tensile strength". Also, the title of the paper implies that they are proposing a solution for all types of materials. While I did not find their work, a comprehensive research work for such a claim. To prove what they claimed they need to show results of several different families of "known" materials, including polymers, ceramics, and metals. This work can be a preliminary test for further investigation.
Response: Agree, thanks for the suggestion.
This manuscript only focuses on ME technology, and the research material is polylactic acid (PLA). The author introduced it in the 1st paragraph in Section 2.2. To further clarify it, the author revised this part according to the reviewer’s comment (see the highlighted content and added reference [23]). The author also changed the title of the article to “Experimental study on improving the mechanical property of material extrusion rapid prototyping polylactic acid parts by applied vibration”. Moreover, the corresponding information is also added in Section 2.3, emphasizing the material of polylactic acid.
Theoretically speaking, adopted in a suitable way, the method (using vibration in forming process) can be applied to the other additive manufacturing techniques and other materials according to the principle of similarity, as both ME technique and PLA material are ordinary. For example, Foroozmehr pioneered introducing vibration (onto the platform) into the manufacturing process, setting up a vibrating laser powder deposition (LPD) equipment. The test results showed that applying vibration could effectively reduce the defect of pores in the LPD parts by 80%, and thereby improve the built parts’ forming quality. The authors confess that this work is a preliminary test for further investigation. More work needs to be done to confirm this conclusion.
- there is a lack of literature review that needs to be done for this work.
Response: Agree, thanks for the suggestion.
Vibration is not widely used in machining field, and it is much less in additive manufacturing field. The authors have revised the part of the literature review, adding 2 relevant literatures published by the authors [19, 20] in this area in Section 1, please have a check the highlighted content.
The above shows the author’s response to the comments and suggestions from the reviewer.
Best Regards,
Shijie

Round 2
Reviewer 2 Report
There have been some improvements in this version.
I recommend deleting Fig 4.
Author Response
Reviewers 2:
There have been some improvements in this version.
I recommend deleting Fig 4.
Response: Thanks for the suggestion and comment.
The author has deleted Figure 4, and revised the corresponding content.
